# The Potential Implications of Hydrogen Sulfide in Aging and Age-Related Diseases through the Lens of Mitohormesis

**DOI:** 10.3390/antiox11081619

**Published:** 2022-08-20

**Authors:** Thi Thuy Tien Vo, Thao Duy Huynh, Ching-Shuen Wang, Kuei-Hung Lai, Zih-Chan Lin, Wei-Ning Lin, Yuh-Lien Chen, Tzu-Yu Peng, Ho-Cheng Wu, I-Ta Lee

**Affiliations:** 1School of Dentistry, College of Oral Medicine, Taipei Medical University, Taipei 11031, Taiwan; 2Lab of Biomaterial, Department of Histology, Embryology, and Genetics, Pham Ngoc Thach University of Medicine, Ho Chi Minh City 72500, Vietnam; 3PhD Program in Clinical Drug Development of Herbal Medicine, College of Pharmacy, Taipei Medical University, Taipei 11031, Taiwan; 4Department of Nursing, Division of Basic Medical Sciences, and Chronic Diseases and Health Promotion Research Center, Chang Gung University of Science and Technology, Puzi City, Chiayi County 61363, Taiwan; 5Graduate Institute of Biomedical and Pharmaceutical Science, Fu Jen Catholic University, New Taipei City 24205, Taiwan; 6Department of Anatomy and Cell Biology, College of Medicine, National Taiwan University, Taipei 10617, Taiwan; 7Graduate Institute of Pharmacognosy, College of Pharmacy, Taipei Medical University, Taipei 11031, Taiwan

**Keywords:** hydrogen sulfide, mitohormesis, mitochondrial unfolded protein response, mitochondrial dynamics, mitophagy

## Abstract

The growing increases in the global life expectancy and the incidence of chronic diseases as a direct consequence have highlighted a demand to develop effective strategies for promoting the health of the aging population. Understanding conserved mechanisms of aging across species is believed helpful for the development of approaches to delay the progression of aging and the onset of age-related diseases. Mitochondrial hormesis (or mitohormesis), which can be defined as an evolutionary-based adaptive response to low-level stress, is emerging as a promising paradigm in the field of anti-aging. Depending on the severity of the perceived stress, there are varying levels of hormetic response existing in the mitochondria called mitochondrial stress response. Hydrogen sulfide (H_2_S) is a volatile, flammable, and toxic gas, with a characteristic odor of rotten eggs. However, H_2_S is now recognized an important gaseous signaling molecule to both physiology and pathophysiology in biological systems. Recent studies that elucidate the importance of H_2_S as a therapeutic molecule has suggested its protective effects beyond the traditional understanding of its antioxidant properties. H_2_S can also be crucial for the activation of mitochondrial stress response, postulating a potential mechanism for combating aging and age-related diseases. Therefore, this review focuses on highlighting the involvement of H_2_S and its sulfur-containing derivatives in the induction of mitochondrial stress response, suggesting a novel possibility of mitohormesis through which this gaseous signaling molecule may promote the healthspan and lifespan of an organism.

## 1. Introduction

In ancient times, the old-age-as-a disease concept was proposed by many scholars. Currently, however, aging and diseases are considered essentially different in nature [1]. Aging can be now defined as an irreversibly progressive decline in the physiological function, which is increasingly recognized as a dominant driving factor for many diseases and conditions that limit human healthspan [2]. An analysis of the Global Burden of Disease Study 2017 reported that 92 diseases, broadly categorized as cardiovascular diseases, chronic respiratory diseases, communicable, maternal, neonatal, and nutritional diseases, diabetes and kidney diseases, digestive diseases, injuries, neoplasms, neurological disorders, and other non-communicable diseases, were identified as age-related, accounting for 51.3% of all global burden among adults [3]. Although human life expectancy has significantly increased over time, this has not been accompanied by a comparable increase in our healthspan [2]. From the perspective of improving the health status and reducing the disease burden of aging populations, it is important to identify the key drivers and understand the mechanisms of aging in order to develop effective strategies for a healthy aging.

Research on the aging process in yeast, nematodes, flies, mice, and humans has suggested that aging is regulated by common and conserved mechanisms in many, if not all species [4,5]. In the other words, understanding the conserved mechanisms in the distantly related model organisms may shed a light on the progression of aging and the onset of age-related diseases in humans. One appealing cause of aging can be traced back to an event over 1.5 billion years ago, which involved the entry of an immortal prokaryote into a more complex but mortal eukaryote, followed by the evolution of the prokaryote into what we now call the mitochondria [6]. Mitochondria are vital organelles in eukaryotic cells, prominent for their crucial roles in energy production through the process of oxidative phosphorylation and in many other cellular processes, such as metabolisms, reactive oxygen species (ROS) generation, calcium homeostasis, apoptosis, and many more [7]. There is increasing evidence to indicate that a decline in the mitochondrial quality and activity is intricately linked to aging, contributing to the development of a wide range of age-related diseases [8,9,10]. The connection between mitochondria and aging has been widely explained by the free radical theory of aging originally proposed by Denham Harman in 1956, which hypothesized that the accumulation of oxidative damage generated by ROS is a key driver of aging [11]. This theory has implied that limiting or inhibiting the generation and accumulation of free radicals, particularly ROS, can reduce the rate of aging, thereby preventing or relieving age-related diseases [12,13]. However, a growing body of study has contradicted this theory [14], suggesting that the relationship between mitochondria and aging is far more complicated. To reconcile the free radical theory of aging, ROS have also been considered as a signaling messenger that may induce protective and adaptive responses [14]. The concept of hormesis, which refers to as evolutionarily adaptive responses of biological systems to moderate the stressors for improving their functionality and tolerance to more severe challenges, has been considered crucial to the resilience and performance, such as for aging and longevity, of a living system [15]. The framework of hormesis can be simply described as low-dose stimulation and high-dose inhibition [16]. In light of mitochondrial basis of aging, mitochondrial hormesis, or mitohormesis, is a biological response through which the induction of mild or moderate mitochondrial stress can promote the viability and wellness of a cell, tissue, or organism. The activation of mitohormetic responses can increase the lifespan in different model organisms, from worms to mammals [17]. In addition, although mitohormesis has been mostly investigated in worms and yeast, a recent study found that this phenomenon can also operate in mammalian macrophages for lipopolysaccharide tolerance, suggesting its important role in the healthspan of higher organisms [18]. Therefore, it is possible to presume that instead of targeting the mitochondria with antioxidants, a strategy that has had low performance or even failure in clinical trials so far [19], harnessing mitohormesis can be a promise in the challenging field of anti-aging, thereby reducing age-related diseases and promoting healthspan.

Hydrogen sulfide (H_2_S) is a colorless, volatile, flammable, and toxic gas, with a characteristic odor of rotten eggs that is detectable by the human nose. The reputation of H_2_S, however, has undergone a paradigm shift with its introduction into the family of gasotransmitters or gaseous signaling molecules alongside nitric oxide (NO) and carbon monoxide (CO), which is endogenously produced in mammals, including humans [20]. Recently, endogenous H_2_S and its sulfur-containing derivatives have been recognized as crucial to many biochemical pathways that regulate the redox homeostasis and mitochondrial function [21]. The literature has indicated that H_2_S possesses antioxidant activities to inhibit free-radical reactions and oxidative damage, providing many benefits in aging and age-related diseases [20,22,23]. Nonetheless, it was reported that the increase in antioxidant protection may shorten the lifespan in worms, namely *Caenorhabditis elegans* (*C. elegans*), whereas the decrease may prolong their lifespan [24]. These findings suggest that the role of oxidative damage in aging remains unclear and that alternative assumptions cannot be ruled out. The concept of mitohormesis, in which high levels of mitochondrial stress cause the cellular damage to promote aging and age-related diseases, whereas a reduced level may rather induce an adaptive response to improve the defense mechanisms, has emerged as a novel anti-aging paradigm. The available data thus far have indicated that redox and stress signaling mechanisms play a pervasive role in mitohormesis, in which ROS are of the utmost importance [17,25]. However, it is possible to re-think the central position of ROS in mitohormesis, since much of the evolutionary biology of life evolved in a sulfur-rich, but virtually oxygen-free atmosphere [21]. Dietary restriction (DR) without malnutrition may provide some benefits in longevity and stress resistance in many organisms across evolutionary boundaries [26], and mitohormesis is considered a potential hypothesis for DR benefits [27]. Interestingly, H_2_S production has been observed in yeast [28], worms [29], fruit flies [30], and rodents [31] in the models of DR-mediated longevity and/or stress resistance, suggesting H_2_S as a novel mediator of DR-induced mitohormetic benefits [32]. These facts may lead to a renewed interest in the sulfide chemistry and biology as another critical element in aging and age-related diseases through the lens of mitohormesis. Therefore, in the present review, we briefly recall the antioxidant activities of H_2_S as a conventional paradigm for preventing aging and age-related diseases. The focus of this review is to propose the involvement of H_2_S and its sulfur-containing derivatives in the induction of hormetic responses in mitochondria, suggesting a novel possibility through which this gaseous signaling molecule may help an organism to respond to stressors and stay healthy during chronological aging.

## 2. Antioxidant Activities: A Conventional Anti-Aging Paradigm of H_2_S

Sulfur is an essential element for living organisms, and sulfur chemistry and biology are, therefore, important to be aware of [33]. H_2_S was well known as an environmental pollutant and a toxic gas until it was discovered as the third endogenous gaseous signaling molecule together with NO and CO [20]. Mammalian cells primarily rely on four enzymatic pathways for the biosynthesis of H_2_S, including cystathionine β-synthase (CBS), cystathionine γ-lyase (CSE), 3-mercaptopyruvate sulfurtransferase (3MST) coupled with cysteine aminotransferase (CAT), and 3MST coupled with d-amino acid oxidase (DAO). Among the enzymes, CBS and CSE are located in the cytosol, whereas 3MST mainly resides in the mitochondria for H_2_S production. However, CBS and CSE can also translocate into the mitochondria when the activity of 3MST is suppressed under challenging conditions, such as hypoxia, to promote the mitochondrial generation of H_2_S [5,34]. The endogenous H_2_S is generally maintained at a low steady-state level to elicit various physiological effects and maintain multiple cellular functions [35], whose decline has been demonstrated to associate to aging and many age-related diseases. It was reported that H_2_S levels in the heart, liver, and kidney tissues were significantly decreased in senescent mice [36]. An observational study indicated a pronounced decrease in plasma levels of H_2_S in African American type 2 diabetic patients as compared with age- and race-matched healthy controls [37]. A reduction in endogenous H_2_S production may also undermine the neurological health as it was found that the levels of H_2_S were substantially lower in the brains of patients with Alzheimer’s disease than in those of age-matched control subjects [38]. A previous review summarized the protective effects of H_2_S against almost all aging hallmarks, except for telomere attrition [5]. In addition, a series of review has highlighted this gaseous molecule as a potential preventive and therapeutic agent in aging and age-related diseases [20,39]; some of them focused on the effects of H_2_S on tissue-specific aging and its implications in associated diseases, particularly the cardiovascular system [40] and central nervous system [23,41,42]. Furthermore, there is a growing body of in vivo evidence to highlight the protective effects of H_2_S on the aging-associated changes in the kidney [43,44]. In a nutshell, oxidative stress caused by excess production and accumulation of ROS is a primary culprit of aging that in turn has a progressive role in age-related diseases. H_2_S at low levels may function as an antioxidant to protect the cells and tissues from oxidative stress primarily by scavenging ROS, reducing the aggregation of lipid peroxidation products, and increasing the generation of intracellular antioxidants, such as superoxide dismutase (SOD), catalase (CAT), glutathione peroxidase (GPx), and glutathione (GSH) (Figure 1). Over the past decades, it has been documented that sulfur-based therapies might play an important role in antioxidant strategies against oxidative damage, which may associate with aging and age-related diseases in humans. It was found that a 3-week therapy with sulfur baths led to a decline in peroxide concentrations and an improvement in total antioxidant status and cholesterol levels in patients with degenerative osteoarthritis [45]. Another study indicated that sulfur baths markedly reduced the blood levels of homocysteine, a risk factor for cardiovascular diseases, in patients with degenerative osteoarthrosis. Nevertheless, there was no significant effects of sulfur baths for the urinary levels of 8-hydroxy-2′-deoxyguanosine (8-OHdG), an indicator of oxidative stress [46]. Hydropinic therapies which involve the drinking of sulfur-containing water may also be beneficial for the enhancement of antioxidant status, possibly furnishing the protection against oxidative damage. An improved body redox status in healthy volunteers was observed after a standard hydropinic treatment with 500 mL sulfurous drinking water per day for 2 weeks. In particular, there was a significant decrease of lipid and protein oxidation products and a substantial increment in the total antioxidant capacity and total thiol levels in the plasma samples from the study group as compared to the control group [47]. Some recent reviews of available H_2_S donors that serve as a platform for the release of H_2_S have underlined their considerable promise as favorable drugs in the clinical treatment of many diseases [48,49,50]. However, a series of experiment on numerous types of animals, including lamprey, trout, mouse, rat, pig, and cow, reported an essentially undetectable basal level of H_2_S gas (<100 nM total sulfide) in blood in all animals. Moreover, exogenous sulfide was also rapidly removed from blood and plasma in real time and in vivo [51]. A very low physiological concentration and prompt turnover suggest that H_2_S may only have a minor importance as a direct antioxidant in mammals, including humans. It is well established that the transcription factor nuclear factor erythroid 2-related factor (Nrf2) is a master regulator of cellular redox homeostasis, whose activity may decline with age. Analyses of Nrf2 signaling indicate a “dedepression” regulatory mechanism, in which Nrf2 is suppressed under a basal condition through Kelch-like erythroid cell-derived protein with CNC homology-associated protein 1 (Keap1)-dependent degradation and it is activated via the modification of critical cysteine thiols of Nrf2 and Keap1. Upon activation, Nrf2 dissociates from Keap1, translocating from the cytoplasm to the nucleus to bind to the antioxidant response element and subsequently induce the expression of multiple detoxification and antioxidant enzymes. As Nrf2 transcriptionally upregulates various genes against oxidative stress, a reduction in Nrf2 function may allow oxidative stress to progress, promoting the aging process [52,53]. Therefore, enhancement of cellular antioxidant defense via the upregulation of Nrf2 pathway may help to maintain the redox balance, mitigating the effects of aging and combating age-related diseases. Interestingly, H_2_S has also been found to alleviate oxidative stress and maintain redox homeostasis through its ability to upregulate the cellular antioxidant mechanisms in an Nrf2-dependent manner [54]. Increasing evidence suggests that modifying thiol groups of cysteine residues in target proteins via S-sulfhydration or persulfidation is an important post-translational modification by H_2_S that is responsible for the biological functions of this gaseous signaling molecule [55,56]. Some animal studies have demonstrated that H_2_S donors that release H_2_S can S-sulfhydrate Keap1 at cysteine-151, leading to Nrf2 dissociation from Keap1 and promoting Nrf2 nuclear translocation, ultimately inducing the expression of downstream protective genes [57,58,59]. These data collectively provide an indirect antioxidant mechanism through which H_2_S signaling may prevent aging and diseases induced by oxidative stress. Nonetheless, a recent study highlighted that H_2_S can stimulate the Fenton reaction, suggesting that it may be a pro-oxidant instead of being an antioxidant [60]. Therefore, it is critical to explore alternative modes of action, apart from antioxidant paradigm, through which H_2_S can encourage healthy aging and prevent age-related diseases. A better understanding of biological activities of H_2_S would be beneficial for the translation of sulfur-based therapies into the clinical practice.

## 3. Mitohormesis: A New Anti-Aging Possibility of H_2_S

The organisms we observe today is a snapshot of a historical experience of preferred life forms that have survived over the past evolutionary stresses to be compatible with the environment and fit for purpose. Therefore, the mechanisms that enable a biological organism to adapt or cope with stresses are important for resilience and performance. The concept of hormesis describes an adaptive response to mild stressors to confer a protection against higher levels of stress in the future, providing a universal and conservative defense mechanism [21]. A recent meta-analysis provided strong evidence for the anti-aging effect of hormesis on *C. elegans* by increasing resistance, promoting healthspan, and prolonging lifespan, postulating a theoretical possibility of hormesis to delay the intrinsic aging in humans [61]. A growing body of research employing various model organisms and human cell cultures has also suggested hormesis as a pro-healthy aging intervention [17,19,62]. Further studies are, therefore, required to determine to what extent the applicability of hormetic effects can be translated from model organisms to human beings.

Mitochondria are crucial to both physiology and pathophysiology, since their roles go beyond being the powerhouse of eukaryotic cells [63]. Literature has reported that mitochondrial function is declined with aging in humans [64], suggesting a connection between mitochondria with aging and age-related diseases [8,9,10]. A vicious cycle in which the increased ROS production induces mitochondrial dysfunction and vice versa is the basis of the free radical theory of aging proposed by Harman [11]. In conflict with Harman’s theory, it is now recognized that ROS may also play a key role in promoting healthspan and lifespan [14,19]. A prevailing hypothesis is that an increased generation of ROS within the mitochondria may lead to an adaptive response that culminates in the improved stress resistance, known as mitochondrial hormesis or mitohormesis [19]. Depending on the severity of the perceived stress, there are three major lines of hormetic response existing in the mitochondria, called the mitochondrial stress response, to adapt to stressful situations. The activation of the mitochondrial unfolded protein response acts as the first line of defense for relieving the stress. However, when the stress level is overwhelming, mitochondrial dynamics and mitochondrial autophagy (mitophagy) come into action for the removal of a part of the mitochondria or the entire damaged mitochondria, respectively [9,10]. It has been proposed that any defect in how the mitochondria sense and respond to a stress is critical for the initiation and progression of aging and age-related diseases [65,66], highlighting the potential implications of mitohormesis in maintaining mitochondrial function and promoting healthy aging.

Oxygen was essentially absent from the atmosphere at the onset of life, and the first eukaryotes appeared and developed in an anoxic and sulfidic environment. As the fundamental principles of regulation have been conserved from simple to complicated organisms, this fact may explain why many regulatory pathways are connected to sulfur biology [21]. Furthermore, oxygen and sulfur are chemically similar: both are chalcogens with six valence electrons. Nonetheless, the electrons of sulfur are farther from the positive nucleus and more favorable for electron transfer reactions than those of oxygen, making it more reactive and versatile [67]. In biological systems, H_2_S and its sulfur-containing derivatives, similar to ROS, are also required for various biochemical pathways that regulate redox homeostasis and mitochondrial function [21]. However, from the perspective of mitohormesis as an anti-aging paradigm, most of the currently available studies have emphasized the biological effects of ROS in mitochondrial stress response [17,19,25], but overlooked those of H_2_S and derivatives. The endosymbiotic hypothesis of mitochondrial origin implies that one of the major targets of H_2_S signaling in eukaryote is the mitochondria [68,69]. Therefore, the importance of this gaseous signaling molecule in mitohormesis should be kept in mind. Nonetheless, a second messenger that transduces H_2_S signal remains unknown. A prevailing hypothesis for H_2_S-mediated signaling may involve the protein persulfidation, which generates highly reactive persulfide species as products [70]. Thus, it is possible to consider that H_2_S in its primary biochemical state alone cannot account for all of the biological effects of this gaseous signaling molecule. Herein, we offer up a new possibility that an extent of mitochondrial stress response may be attributed to H_2_S and intermediates or derivatives of sulfide metabolites that are produced through the oxidation of H_2_S (Figure 2).

### 3.1. Mitochondrial Unfolded Protein Response

The mitochondrial unfolded protein response (UPR^mt^) is a stress response pathway that restores the mitochondrial homeostasis, particularly proteostasis. Mitochondrial protein homeostasis is maintained through a proper folding and assembly of newly translated polypeptides as well as an efficient trafficking and turnover of those proteins that fail to fold correctly. In response to the accumulation of unfolded or misfolded proteins exceeding the chaperone protein-folding capacity of mitochondria, UPR^mt^ is activated to induce a retrograde signaling pathway from the mitochondria to the nucleus, leading to the expression of proteases, chaperonins, and many stress response genes to degrade the unfolded or misfolded proteins, increase the chaperone capacity, and re-establish the mitochondrial protein homeostasis [71]. Literature has also indicated that a disruption of mitochondrial proteostasis plays an important role in the UPR^mt^ activation that has crucial implications in aging [10]. For example, an increased protein-folding workload in the mitochondria as a proteostatic stress may trigger the UPR^mt^ activation in *C. elegans* [72]. Suggestive of potential relevance in mammals, the UPR^mt^ network is also active in various tissues [73], which is deregulated in age-related diseases such as sarcopenia (muscle wasting and weakness) [74] and Alzheimer’s disease [75,76]. In addition, mitochondrial stresses can lead to the production of stress response molecules termed mitokines; some of them are expressed in response to UPR^mt^, such as fibroblast growth factor 21 (FGF21) and growth differentiation factor 15 (GDF15) [10]. Both mitokines have shown many beneficial effects, including cardioprotective, neuroprotective, metabolic, anti-inflammatory, and anti-immunosenescent activities [77,78,79,80]. In an observational study, the plasma levels of FGF21 and GDF15 were significantly associated with aging and reached maximal in centenarians, suggesting that the biological mechanisms triggering their production remain active in the extreme old age. However, their levels were negatively correlated with parameters indicative of health status and survival. Interpreted in the framework of mitohormesis, these mitokines may represent an attempt of the cells or tissues and the whole organism to cope with stressors, whose efficiency is dependent on the intensity of the stress [81]. Taken together, the recovery of proteostasis via the UPR^mt^ activation can be a conserved mechanism that underlines the beneficial effects of mitohormesis in aging and age-related diseases. Interestingly, treatment with H_2_S was proved to protect *C. elegans* from hypoxia-induced disruption of proteostasis [82]. Similarly, a significant increase in protein aggregation accompanied by a lower level of brain thiol groups and a decreased expression of brain CBS were observed in the forebrain of Zucker diabetic fatty rats. Treatment with a sodium salt of HS− (i.e., deprotonated H_2_S) named sodium hydrosulfide (NaHS), a commonly used H_2_S donor, normalized the increased protein synthesis and aggregation in the cultured brain tissues [83]. Furthermore, in mammals, there is a sulfide oxidation pathway that resides in the mitochondria, whose first and committing step is catalyzed by the sulfide quinone oxidoreductase (SQRD), which is a highly conserved mitochondrial enzyme. The SQRD prevents H_2_S accumulation and generates highly reactive persulfide species as products, which can be further oxidized or can modify cysteine residues in proteins via persulfidation [70]. A past study demonstrated that UPR^mt^ is activated in *sqrd-1* (an orthologue of the conserved SQRD) mutant *C. elegans* exposed to H_2_S but not in wild-type animals, suggesting that H_2_S may coordinate this mitochondrial stress response and promote proteostasis [84]. In addition, it was reported that the knock-down of SQRD in Hepa1-6 cells prior to stimulated ischemia reperfusion injury had no significant impact on the protective effects of exogenous H_2_S, but partially inhibited the protection of thiosulfate as a product of H_2_S oxidation via SQRD, implying the importance of other sulfide metabolites in H_2_S-mediated stress resistance [32]. With these in mind, we presume that H_2_S signaling can regulate some aspects of UPR^mt^ to reduce the age-associated decline in mitochondrial proteostasis, thereby promoting healthy aging and preventing age-related diseases. 

### 3.2. Mitochondrial Dynamics

Mitochondria are highly dynamic organelles that undergo coordinated cycles of fission and fusion, referred to as mitochondrial dynamics, to maintain their size, shape, and distribution. Mitochondrial fission is characterized by the division of one mitochondrion into two daughter mitochondria, which is regulated by the recruitment of the GTPase Dynamin-related protein 1 (Drp1) and Drp2. In contrast, mitochondrial fusion is the union of two mitochondria into one mitochondrion, which is driven by a two-step process with the outer mitochondrial membrane fusion mediated by mitofusins 1 (Mfn1) and Mfn2 followed by the inner membrane fusion mediated by optic atrophy 1 (OA1) [85]. Mitochondrial dynamics is an essential mitochondrial stress response to ensure cellular homeostasis and survival upon stress. The disruption in any of these events is linked to aging and age-related diseases [10]. In fact, fragmentation of the mitochondrial network is increased with aging in *C. elegans* [86]. Similarly, aging is associated with a higher content of Drp1 in the mitochondrial fraction of aged Fischer 344xBN animals as compared to young animals [87]. In contrast, a reduction in mitochondrial fission by the deletion of the homolog of mammalian *drp-1* can increase the lifespan and fitness in fungal aging models [88]. Interestingly, an in vitro treatment with NaHS demonstrated that H_2_S reduced Drp1 expression and inhibited mitochondrial fission in neuroblastoma cells, suggesting it as a promising candidate for diseases mediated by mitochondrial dynamics such as Alzheimer’s disease [89]. Moreover, it was found that Drp1 activity was negatively regulated by persulfidation [90], a process through which H_2_S can signal by attaching a thiol to a reactive and accessible cysteine of a protein to alter the protein function [69]. A recent review further proposed a crosstalk between the mitochondrial fission/fusion cycle and reactive sulfur species (RSS)-dependent Drp1 activation, providing a new insight about the actions of RSS in cardiac senescence [91]. Since H_2_S contributes a part to the bioavailability of RSS, including thiyl radical (HS•), hydrogen persulfide (H_2_S_2_), persulfide supersulfide radical (HS_2_•−), and elemental sulfur, in biological systems [67], there is a possibility of H_2_S to prevent the stressed mitochondria from undergoing maladaptive dynamics via RSS production. However, maintaining the balance between mitochondrial fission and fusion should be a more promising strategy for the management of aging and age-related diseases than stimulating or inhibiting either of the two events. Therefore, the future research models should include both mechanisms instead of targeting fusion or fission alone to better understand the effects of H_2_S and its derivatives on this mitochondrial stress response. When our knowledge of the importance of sulfide biology in mitochondrial quality control is evolved, H_2_S-based therapeutics for the manipulation of mitochondrial dynamics might become a powerful tool to promote healthy aging and prevent age-related diseases.

### 3.3. Mitophagy

Autophagy refers to lysosomal mediated degradation processes, providing a crucial mechanism for the recycling and clearance of excess or damaged cellular constituents and organelles [92]. A study investigating Schwann cells isolated from the nerves of 2-day-old and 25-month-old rats for their autophagic capacity found a slowed mobility of lysosomal-associated membrane protein 1 (LAMP1) and a reduced co-localization of LAMP1 with microtubule-associated protein 1 light chain 3 (LC3), two key regulators of autophagy, suggesting an altered autophagy-lysosomal activity of Schwann cells isolated from older rats, indicative of defective autophagy in aged animals. Interestingly, it was demonstrated that the young cells can respond to DR by activating the autophagic pathway, as evidenced by an increase in the level of autophagy-related (Atg)7 and the conversion of LC3-I to its lipidated form LC3-II as compared to the old cells [93]. Literature has reported the H_2_S production in various models of DR-mediated longevity and/or stress resistance [28,29,30,31], suggesting that H_2_S might be involved in DR-induced benefits such as autophagy. In fact, it was reported that diallyl trisulfide, a fully characterized H_2_S donor, can induce autophagy in human urothelial carcinoma cells, as evidenced by the increased expression of many important autophagic genes, such as Atg13, Becn1, Lamp1, and Unc-51 like autophagy activating kinase (Ulk)1. In addition, Western blot analysis found that this H_2_S donor significantly enhanced the expression of LC3-II protein, which is considered one of critical events for autophagosome formation and activity [94]. Consistently, an animal study showed that the treatment with NaHS markedly recovered the impaired locomotor activity and ameliorated the inflammatory/fibrotic status in dystrophic mice as compared to control mice, which was in part attributed to the recovery in the expression of key genes regulating autophagy, including Lamp1, Becn1, phosphatidylinositol 3-kinase catalytic subunit type 3 (PiK3c3), Atg3, Atg4, Atg7, and Ulk1. It was further demonstrated that the lipidation of LC3-I into LC3-II was compromised in dystrophic mice, but was rescued by treatment with NaHS [95]. This evidence, thus, proposes that the H_2_S signaling pathway may represent a novel strategy for regulating autophagy. However, it is important to keep in mind that autophagy is no longer considered a non-selective bulk degradation pathway [96]. From the perspective of mitohormesis, when the mitochondrial stress accumulates to a level that overwhelms the capacity of stress response, a selective type of autophagy targeting the aged and/or damaged mitochondria for degradation, termed mitophagy, takes place. Among all mitochondrial stress responses, mitophagy is the only mechanism that regulates the turnover of the whole organelle, reinstating cellular homeostasis and maintaining cellular fitness under conditions of stress. It generally operates through ubiquitin-dependent and ubiquitin-independent pathways; both of them culminate in the engulfment of mitochondria by autophagosomes. The ubiquitin-dependent mitophagy is regulated by the phosphatase and tensin homolog (PTEN)-induced kinase 1 (PINK1)/cytosolic E3 ubiquitin ligase (Parkin) axis, which ubiquitinates various substrates to recruit autophagy receptors, whereas the ubiquitin-independent mechanism is driven by a direct recruitment of autophagy receptors [10,97]. There is increasing evidence to indicate that mitophagy affects aging and lifespan in a wide variety of organisms. For instance, an accrual of mitochondria with age was observed in wild-type *C. elegans*, and the depletion of two autophagy regulator genes *dct-1* and *bec-1* was found to impair the removal of mitochondria to lead to a progressive accumulation of mitochondria during aging in the nematodes. Moreover, the deficiency in PINK1 and Parkin, two critical conserved components of mitophagy, suppressed mitophagy, compromised mitochondrial morphology, and destabilized mitochondrial network in wild-type *C. elegans* cells under stress conditions [98]. It was reported that Drosophila parkin null mutants exhibited mitochondrial dysfunction and reduced lifespan [99], whereas Parkin upregulation in adult Drosophila melanogaster reduced proteotoxicity and altered mitochondrial dynamics during aging to lead to a more prolonged lifespan [100]. Mitophagy also declines in several tissues, such as brains [101] and hearts [102], in aged mice, contributing to pathological processes. In lines with these findings, an experimental study proved that the beneficial effects of urolithin A (a first-in-class natural compound) on aging and lifespan in *C. elegans* were dependent on mitochondrial function, particularly via the induction of mitophagy [103]. In addition, mitophagy can also be activated in response to hypoxia as an adaptive metabolic response, which is required for reducing ROS levels and preventing cell death [104]. It has been postulated that the manipulation of mitophagy that plays a critical role in mitochondrial quality control by hypoxia or pharmacological approaches may offer a new promise for cardioprotection [105]. Interestingly, there is increasing evidence to demonstrate H_2_S as a mitophagy modulator that may mitigate the effects of aging. A protective effect of H_2_S supplementation using GYY437 as a slow-releasing donor was observed in primary mouse hepatocytes. In particular, the mitochondria showed a high ability to recover and the mitophagy biomarkers were increased two-fold in response to GYY437 under stress conditions [106]. An animal study also found that exogenous H_2_S promoted the Parkin translocation into the mitochondria and enhanced its catalytic activity via sulfhydration, leading to mitophagy in the hearts of mice [107]. Similarly, a major decline in Parkin sulfhydration was observed in the corpus striatum of Parkinson’s disease patients, suggesting that this loss may be pathogenic [108]. Cysteine, which serves as a substrate for the production of H_2_S, has also been documented to upregulate mitophagy [109], further supporting the role of H_2_S in the regulation of mitophagy. Taken together, mitophagy may represent a conserved mechanism to maintain mitochondrial quality, and practical strategies that boost the endogenous synthesis of H_2_S or deliver the exogenous source of this gas into the biological systems may be beneficial for the field of anti-aging. Nevertheless, while emerging evidence has suggested that the activation of mitophagy may elicit pro-survival and pro-healthy responses, it may be linked to cell death if overactivated [66]. Further studies are, therefore, needed to determine to what extent the upregulation of mitophagy is protective or deleterious to a cell, tissue, and organism.

## 4. Conclusions and Future Prospects

The gaseous signaling molecule H_2_S is now considered pivotal to both physiology and pathophysiology across tissues and species. A growing body of research has demonstrated the benefits of H_2_S-based interventions, whether dietary or pharmacological modalities, in promoting healthspan and lifespan, suggesting H_2_S as a potential approach for combating aging and age-related diseases. Even though we have only recently begun to harness H_2_S-based interventions in earnest in clinical practice, there are many aspects of the involvement of H_2_S signaling in aging biology and age-associated changes that remain unclear. Mitochondria are vital organelles in eukaryotic cells, prominent for their important roles in energy production, metabolisms, ROS generation, and many more. Literature has recognized an intricate connection between a decline in the mitochondrial quality with aging and age-related diseases. There is an increasing interest of employing H_2_S as a potent candidate to target mitochondria, providing a promising avenue for controlling mitochondria quality, thereby delaying aging and treating age-related diseases, such as cardiovascular diseases, neurodegenerative diseases, and even cancers. From the perspective of the free radical theory of aging, the efficiency in antioxidant activities of H_2_S may serve as a protective shield against aging and associated conditions. Nevertheless, manipulation of ROS is not as straightforward as it might seem, since our efforts to alleviate the pro-aging effects of ROS with antioxidants has still had limited success. An alternative hypothesis of mitohormesis, in which an increased generation of ROS within the mitochondria leads to an adaptive response that culminates in an improved stress resistance, has aroused the interest as a new paradigm in the field of anti-aging. From the standpoint of mitohormesis, any defect in how the mitochondria sense and respond to stress may contribute to aging and age-related diseases. Therefore, a biologically active molecule that can stimulate mitohormesis may provide a promise of biomedical applicability to anti-aging and a wide variety of age-related diseases. Depending on the severity of the perceived stress, there are varying levels of hormetic response existing in mitochondria called mitochondrial stress response. Interestingly, emerging evidence has suggested that H_2_S can also be important for the activation of mitochondrial stress response, both directly and indirectly, as this gas itself may be a pro-oxidant by stimulating the Fenton reaction or it may undergo the sulfide oxidation pathway in the mitochondria to generate highly reactive species which can be further oxidized or can modify cysteine residues via persulfidation. While the concept of a mild mitochondrial stress being good is attractive, it remains difficult to determine the boundary between beneficial mitochondrial hormesis and deleterious mitochondrial collapse in response to H_2_S and its derivatives. Moreover, considering the key involvement of mitochondria in H_2_S oxidation pathway, further studies are needed to define the delicate balance between H_2_S oxidation and mitochondrial quality control. Although we are still far from a full understanding of how H_2_S operates at the mitochondrial signaling pathways to promote pro-healthy aging in organisms, practical strategies which either boost an endogenous synthesis of H_2_S or which provide an exogenous source of this gas may have a bright future in the challenging field of anti-aging through the lens of mitohormesis. Undoubtedly, a better comprehension of the connection between H_2_S and mitochondria in aging biology and age-related changes would improve our efforts for future interventions. In addition, it is important to keep in mind that excessive levels of H_2_S are very toxic to biological systems. Therefore, further studies should also focus on elucidating other biological aspects, beyond the mitochondria, through which H_2_S operates in living organisms.

## Figures and Tables

**Figure 1 antioxidants-11-01619-f001:**
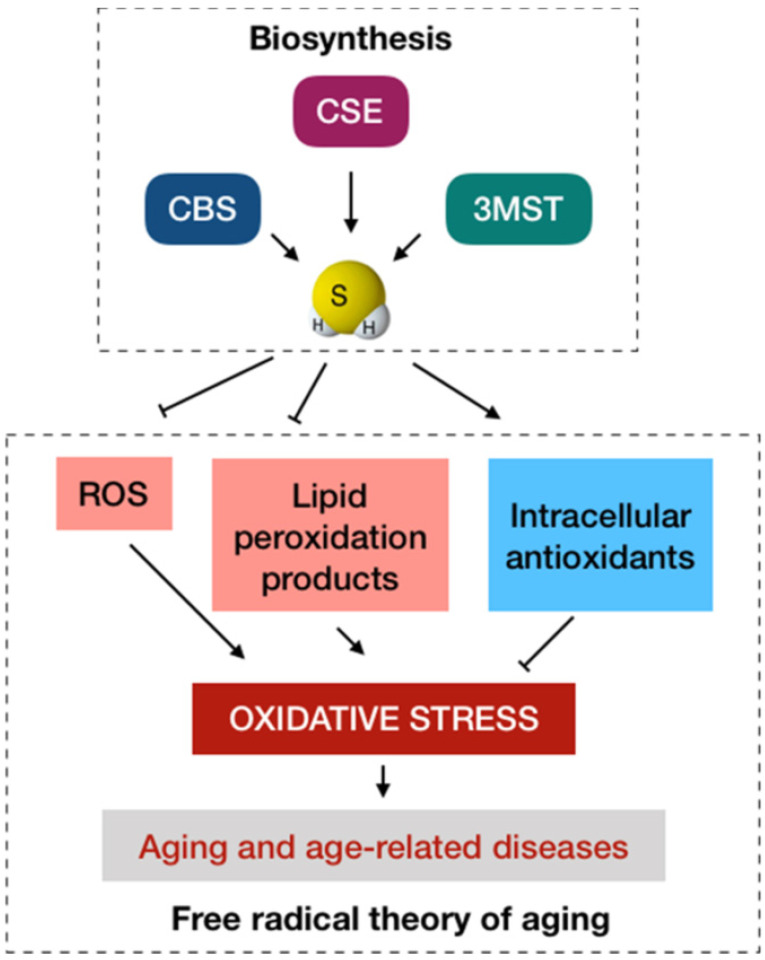
The conventional antioxidant paradigm of H_2_S in anti-aging. The biosynthesis of H_2_S in mammalian cells primarily involve cystathionine β-synthase (CBS), cystathionine γ-lyase (CSE), and 3-mercaptopyruvate sulfurtransferase (3MST). From the viewpoint of the free radical theory of aging, oxidative stress caused by the excessive production and accumulation of ROS is a major cause of aging that in turn has a progressive role in age-related diseases. H_2_S at physiological levels may function as an antioxidant to protect the cells and tissues from oxidative stress by scavenging ROS, reducing the aggregation of lipid peroxidation products, and increasing the generation of intracellular antioxidants.

**Figure 2 antioxidants-11-01619-f002:**
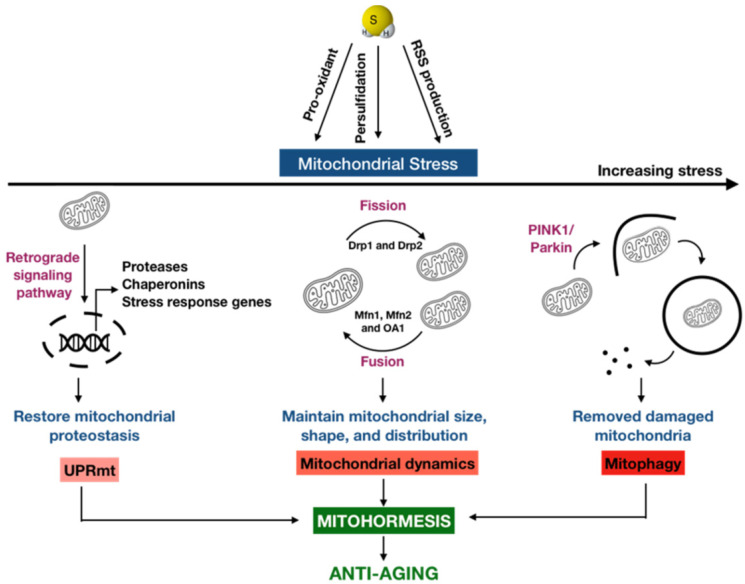
A new possibility of H_2_S in anti-aging via mitohormesis. Mitohormesis refers to as an adaptive response through which the induction of mild or moderate mitochondrial stress can promote the viability and health of a cell, tissue, or organism. Depending on the severity of the perceived stress, there are varying hormetic responses existing in the mitochondria called mitochondrial stress response to adapt to different stress levels. The mitochondrial unfolded protein response (UPR^mt^) is activated as the first line of defense for mitigating the stress, which induces a retrograde signaling pathway from the mitochondria to the nucleus, leading to the expression of proteases, chaperonins, and many stress response genes to degrade the unfolded or misfolded proteins and re-establish the mitochondrial protein homeostasis. In addition, mitochondria are highly dynamic organelles that undergo coordinated cycles of fission and fusion, referred to as mitochondrial dynamics, to maintain their size, shape, and distribution. When the stress level is increased, mitochondrial dynamics comes into action. When the mitochondrial stress level is overwhelmed, a selective type of autophagy targeting the damaged mitochondria for degradation, termed mitophagy, takes place. H_2_S can be an important regulator for these mitochondrial stress responses, both directly and indirectly, as this gas itself may be a pro-oxidant by stimulating the Fenton reaction or it may undergo the sulfide oxidation pathway in the mitochondria to generate highly reactive sulfur species (RSS), which can be further oxidized or can modify cysteine residues via persulfidation.

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
