# Peer review of "The Potential Implications of Hydrogen Sulfide in Aging and Age-Related Diseases through the Lens of Mitohormesis"

_antioxidants, 2022, doi:10.3390/antiox11081619_

Round 1

Reviewer 1 Report

This is a good and comprehensive discussion of H2S and Mitochondrial hormesis crosstalk in aging and age related disease. I just have a few comments: 

1.    Authors aim to discuss antiaging effect of H2S via mitohormesis but relatively little space is devoted to discuss the post-translational modification called s-sulphydration or persulfidation induced by H2S. For example, H2S is known to s-sulphydrilated Keap-1, which interacts with Nrf2, a master regulator of the antioxidant response.

2.    among the mechanisms of mitophagy, there is also the activation of Lamp1, Becn1, PiK3c3, Atg3, Atg4, Atg7, LC3-I and II, Ulk1, the effect of H2S should be also mentioned

Author Response

Manuscript ID: Antioxidants-1859940

Dear Editors and Reviewers,

Thank you very much for your critical comments on our manuscript. You pointed out the remaining issues that require further modifications, thereby improving the scientific content in our manuscript. We have revised our manuscript according to your comments. Please refer to the list of Responses to Comments in the below pages.

We hope that the revised manuscript will be satisfactory for publication in the Antioxidants, and look forward to hearing from you about the acceptability of the paper. Thank you so much for your considerations.

With best regards,

I-Ta Lee, Ph.D.

School of Dentistry, College of Oral Medicine, Taipei Medical University, Taipei, Taiwan

250 Wuxing St. Taipei 11031, Taiwan

Tel: +886-2-27361661 ext. 5162

Fax: +886-2-27362295

E-mail addresses: itlee0128@tmu.edu.tw

Comments and Suggestions for Authors

This is a good and comprehensive discussion of H2S and Mitochondrial hormesis crosstalk in aging and age related disease. I just have a few comments:

  1. Authors aim to discuss antiaging effect of H2S via mitohormesis but relatively little space is devoted to discuss the post-translational modification called s-sulphydration or persulfidation induced by H2S. For example, H2S is known to s-sulphydrilated Keap-1, which interacts with Nrf2, a master regulator of the antioxidant response.

        Responses to Comments:

       Thank you very much for your critical comment. We have discussed the anti-aging effect of H2S via the post-translational modification, i.e. the S-sulfhydration of Keap1, to upregulate the Nrf2 signaling pathway, a master regulator of cellular antioxidant defense (Page 4 & 5, Line 186-210).

  1. Among the mechanisms of mitophagy, there is also the activation of Lamp1, Becn1, PiK3c3, Atg3, Atg4, Atg7, LC3-I and II, Ulk1, the effect of H2S should be also mentioned.

        Responses to Comments:

        Thank you very much for your critical comment. We have discussed the effects of H2S on the induction of key genes regulating autophagy (e.g. Lamp1, Becn1, PiK3c3, Atg3, Atg4, Atg7, LC3-I and II, and Ulk1) to further highlight its importance in this self-protective mechanism (Page 9 & 10, Line 412-448).

Reviewer 2 Report

This review highlights the potential role of hydrogen sulfide and related reactive sulfur species in the induction of mitochondrial stress response. The current literature is appropriately covered and the idea is quite interesting, but remains speculative/suggestive at this time.  More studies are certainly needed to solidify the connections proposed, and perhaps this review will be a catalyst for such work.

Some minor points --

On page 8, line 306:  "H2S precursor named sodium hydrosulfide (NaHS)" -- I would not call this a precursor.  It is simply the sodium salt of HS- (i.e., deprotonated H2S).

At the end of the manuscript, Author Contributions and Funding information needs to be added.

Author Response

Manuscript ID: Antioxidants-1859940

Dear Editors and Reviewers,

Thank you very much for your critical comments on our manuscript. You pointed out the remaining issues that require further modifications, thereby improving the scientific content in our manuscript. We have revised our manuscript according to your comments. Please refer to the list of Responses to Comments in the below pages.

We hope that the revised manuscript will be satisfactory for publication in the Antioxidants, and look forward to hearing from you about the acceptability of the paper. Thank you so much for your considerations.

With best regards,

I-Ta Lee, Ph.D.

School of Dentistry, College of Oral Medicine, Taipei Medical University, Taipei, Taiwan

250 Wuxing St. Taipei 11031, Taiwan

Tel: +886-2-27361661 ext. 5162

Fax: +886-2-27362295

E-mail addresses: itlee0128@tmu.edu.tw

Comments and Suggestions for Authors

This review highlights the potential role of hydrogen sulfide and related reactive sulfur species in the induction of mitochondrial stress response. The current literature is appropriately covered and the idea is quite interesting, but remains speculative/suggestive at this time.  More studies are certainly needed to solidify the connections proposed, and perhaps this review will be a catalyst for such work.

Some minor points --

On page 8, line 306:  "H2S precursor named sodium hydrosulfide (NaHS)" -- I would not call this a precursor.  It is simply the sodium salt of HS- (i.e., deprotonated H2S).

        Responses to Comments:

        Thank you very much for your critical comment. We used “H2S precursor” to interchange “H2S donor”. We have revised “H2S precursor named sodium hydrosulfide (NaHS)” into “sodium salt of HS− (i.e., deprotonated H2S) named sodium hydrosulfide (NaHS), a commonly used H2S donor” for accuracy (Page 8, Line 341-342).

At the end of the manuscript, Author Contributions and Funding information needs to be added.

        Responses to Comments:

        Thank you very much for your critical comment. We have added the Author Contributions and Funding information at the end of the manuscript (Page 12, Line 561-566).
